# Isolation, cryo-laser scanning confocal microscope imaging and cryo-FIB milling of mouse glutamatergic synaptosomes

**Prerana Gogoi[1], Momoko Shiozaki[2], Eric Gouaux [1,3]***

1 Vollum Institute, Oregon Health and Science University, Portland, Oregon, United States of America,
2 Howard Hughes Medical Institute, Janelia Research Campus, Ashburn, Virginia, United States of America,
3 Howard Hughes Medical Institute, Oregon Health and Science University, Portland, Oregon, United States of America

* gouauxe@ohsu.edu

**Data Availability Statement:** All relevant data are within the paper and its Supporting information files.

## Abstract

Ionotropic glutamate receptors (iGluRs) at postsynaptic terminals mediate the majority of fast excitatory neurotransmission in response to release of glutamate from the presynaptic terminal. Obtaining structural information on the molecular organization of iGluRs in their native environment, along with other signaling and scaffolding proteins in the postsynaptic density (PSD), and associated proteins on the presynaptic terminal, would enhance understanding of the molecular basis for excitatory synaptic transmission in normal and in disease states. Cryo-electron tomography (ET) studies of synaptosomes is one attractive vehicle by which to study iGluR-containing excitatory synapses. Here we describe a workflow for the preparation of glutamatergic synaptosomes for cryo-ET studies. We describe the utilization of fluorescent markers for the facile detection of the pre and postsynaptic terminals of glutamatergic synaptosomes using cryo-laser scanning confocal microscope (cryo-LSM). We further provide the details for preparation of lamellae, between ~100 to 200 nm thick, of glutamatergic synaptosomes using cryo-focused ion-beam (FIB) milling. We monitor the lamella preparation using a scanning electron microscope (SEM) and following lamella production, we identify regions for subsequent cryo-ET studies by confocal fluorescent imaging, exploiting the pre and postsynaptic fluorophores.

## Introduction

Glutamate released from the presynaptic terminal acts upon the postsynaptic ionotropic glutamate-receptor ion channels (iGluRs) that include the AMPA ($\alpha$-amino-3-hydroxy-5-methyl-4-isoxazolepropionic acid), NMDA ($N$-methyl-D-aspartic acid) and kainate receptors, causing the influx of cations ($Na^+$, $K^+$ and $Ca^{2+}$) and resulting in excitatory synaptic transmission [1,2]. iGluRs are mostly concentrated in the postsynaptic density (PSD) and are anchored by an intricate web of specialized protein molecules that regulate their trafficking and modulate their expression and functional properties, influencing synaptic plasticity [3–6]. In the "lateral" dimension of a synapse, AMPA receptors (AMPAR) and NMDA receptors (NMDAR) are arranged in a distinctive subsynaptic distribution to align with the presynaptic release site,

**Funding:** This work was supported by the NIH (NINDS) grant 2R01NS038631 to E.G. and E.G. is an investigator with the Howard Hughes Medical Institute. The funders had and will not have a role in study design, data collection and analysis, decision to publish, or preparation of the manuscript.

**Competing interests:** The authors have declared that no competing interests exist.

which in turn, influences receptor activation [7]. Within the synapse, AMPARs and NMDARs are organized into subregions of higher receptor density termed nanodomains or nanoclusters [8–10]. A typical hippocampal synapse contains one to three nanodomains, 80–100 nm in diameter, with an estimated ~25 receptors per nanocluster and ~100 receptors per synapse [9–11]. However, depending on the brain region and synapse size, the number and size of the nanodomain varies [12]. AMPAR nanodomains are localized at the PSD periphery and broadly distributed across the synapse, while NMDAR nanodomains occupy the central region of the PSD. [11,13–17]. A visual insight into the arrangement of the iGluRs in the postsynaptic terminal, in conjunction with the presynaptic terminal, would contribute towards understanding the molecular basis of synaptic transmission.

One attractive model for studying synapses are pinched-off synaptic nerve terminals, known as synaptosomes [18,19]. Typically, synaptosomes are ~0.5–1 μm in diameter and consist of re-sealed presynaptic and postsynaptic nerve terminals with the ability to retain functional properties such as membrane potential and depolarization-induced neurotransmitter release [20–23]. A re-sealed presynaptic compartment encloses the contents of the nerve terminal such as synaptic vesicles, mitochondria and cytoskeleton. The postsynaptic termini within a synaptosome carries a portion of the postsynaptic membrane along with the postsynaptic density (PSD). Most importantly, the postsynaptic membranes bear receptors including iGluRs, along with a set of scaffold proteins that constitute the PSD and hold the receptors in position [24–32]. Density gradient centrifugation using either sucrose, Ficoll or Percoll have been popularly used for isolating synaptosomes. These methods are especially useful for nerve terminals on dendritic spines and their application results in synaptosomes containing all the neurotransmitter types [33–38]. Over time, attempts have been made to reduce the preparation time in order to minimize synaptosomal shrinkage and mechanical damage and to increase viability and functional integrity [37–42].

Synaptosomes can be employed as an experimental system for gaining insight into the structural organization of iGluRs at the PSD using present-day structure determination techniques. Recent developments in the field of cryo-electron tomography (ET) makes it an attractive tool to elucidate the biological structures such as glutamatergic synaptosomes in their near-to-native state [43–46]. Vitreous sectioning of mammalian synapses in organotypic slices or in dissociated primary neuronal cultural have been applied to image synapses using cryo-ET [47–49]. However, vitreous sections suffer from substantial compression artifacts and primary neuronal cultures tend to grow into thick areas which are difficult to image. While a previous study was successful in performing cryo-ET of cultured hippocampal neurons in distinguishing excitatory and inhibitory synapses [50], methods to visualize synapses derived from native brain tissue may allow for additional insights into the structure and organization of synaptic zones. Another recent study demonstrated the advantages of utilizing a synaptosomal preparation [46]. However, no studies of cultured neurons or of synaptosomes have exploited fluorescent markers to unambiguously identify GluA2-containing glutamatergic synapses. To address this issue, we have developed a workflow to prepare artifact-free thin (~100–200 nm) dual fluorescently-labelled glutamatergic synaptosomes on cryo-electron microscopy (cryo-EM) grids by utilizing cryo-focused ion beam (cryo-FIB) milling [51–53]. We utilized a knock-in mouse line that expresses a fully functional fluorescently (mVenus) labelled vesicular glutamate transporter-1 (vGLUT1), a specific presynaptic marker for glutamatergic synapses [54]. For identification of post synapses, we utilized a well characterized GluA2 subunit specific antibody fragment, 15F1 Fab, tagged with mCherry (15F1 Fab-mCherry) [55,56].

Here we detail a workflow for the preparation of glutamatergic synaptosomes using three different methods of density gradient centrifugation for subsequent cryo-ET studies (Fig 1). Irrespective of the density gradient centrifugation method used, the preparation time of

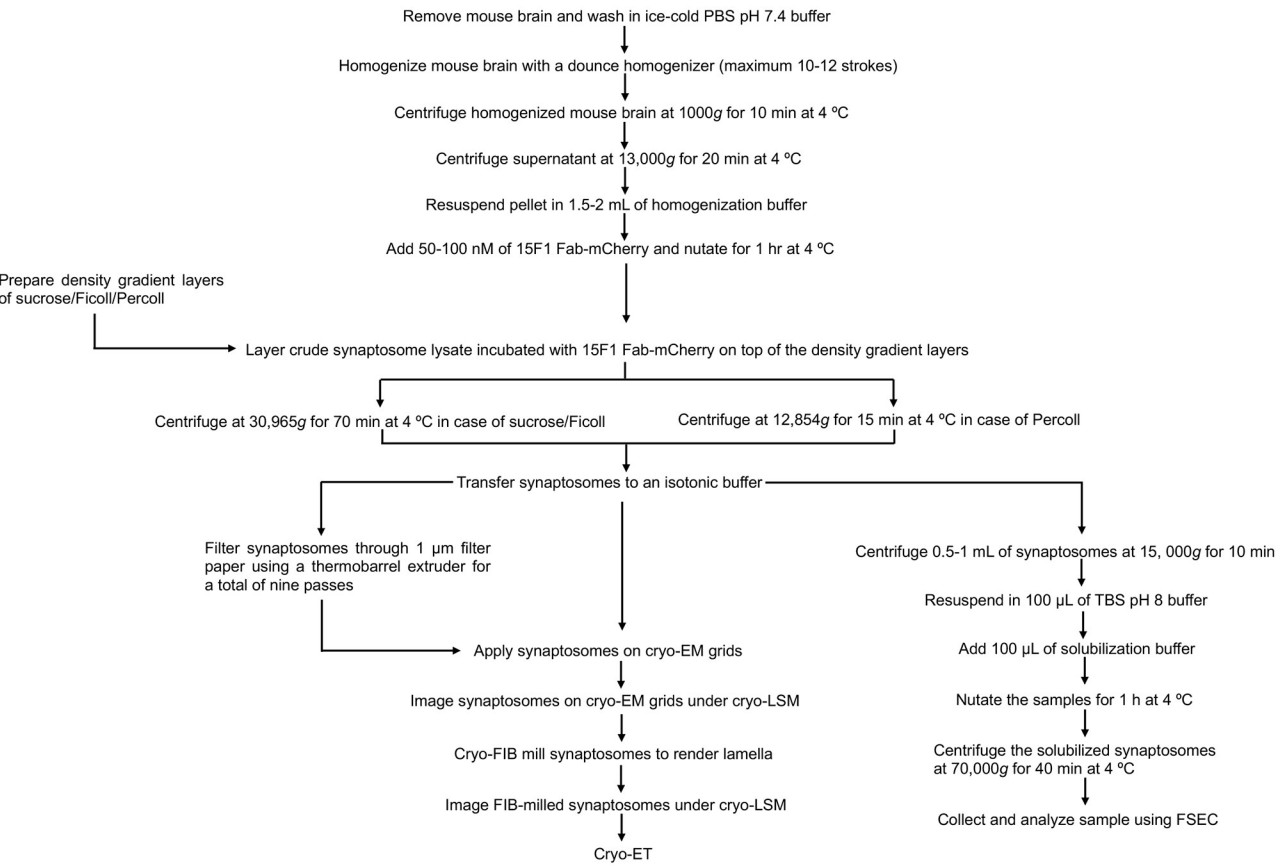

**Fig 1. Workflow for the preparation of cryo-FIB milled glutamatergic synaptosomes for cryo-ET studies.**

synaptosomes to cryo-EM grid preparation can be completed in ~4–5 hrs. Cryo-confocal fluorescence microscopy was employed to identify glutamatergic synaptosomes. Subsequently, fluorescence guided cryo-focused ion beam (cryo-FIB) milling was performed for rendering lamellae suitable for cryo-ET studies.

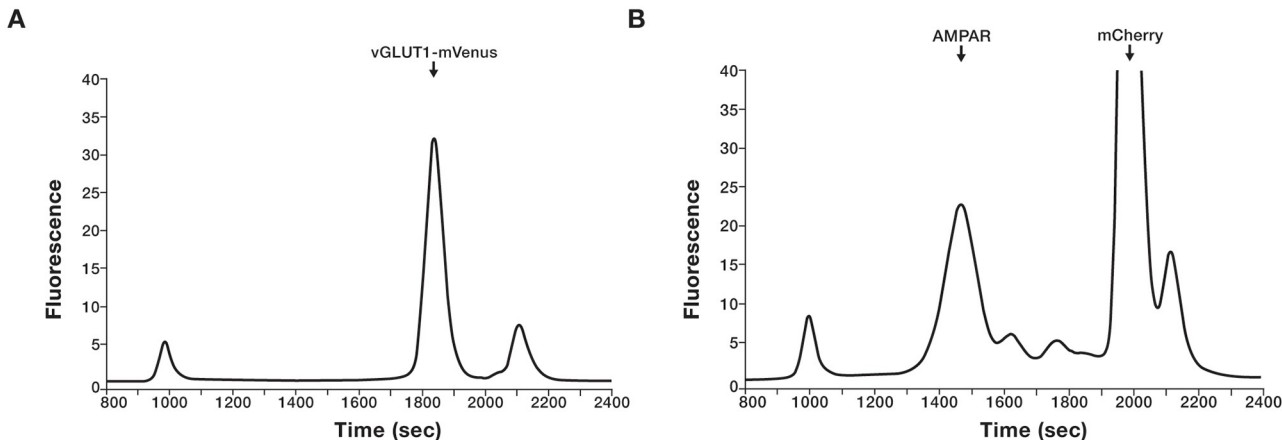

**Fig 2. FSEC analysis of isolated glutamatergic synaptosomes.** Detection of (A) vGLUT1-mVenus and (B) AMPAR bound to 15F1 Fab-mCherry in isolated synaptosomes using Venus (λex: 510 nm, λem: 530 nm) and mCherry (λex: 580 nm, λem: 610 nm) channels, respectively, via FSEC.

## Materials and methods

The protocol described in this peer-reviewed article is published on protocols.io dx.doi.org/10.17504/protocols.io.kxygxz5mkv8j/v1 and is included for printing as S1 File with this article.

## Expected results

We utilized fluorescence-detection size-exclusion chromatography (FSEC) [57] to confirm the presence of glutamatergic synaptosomes in the retrieved fraction after sucrose, Ficoll or Percoll

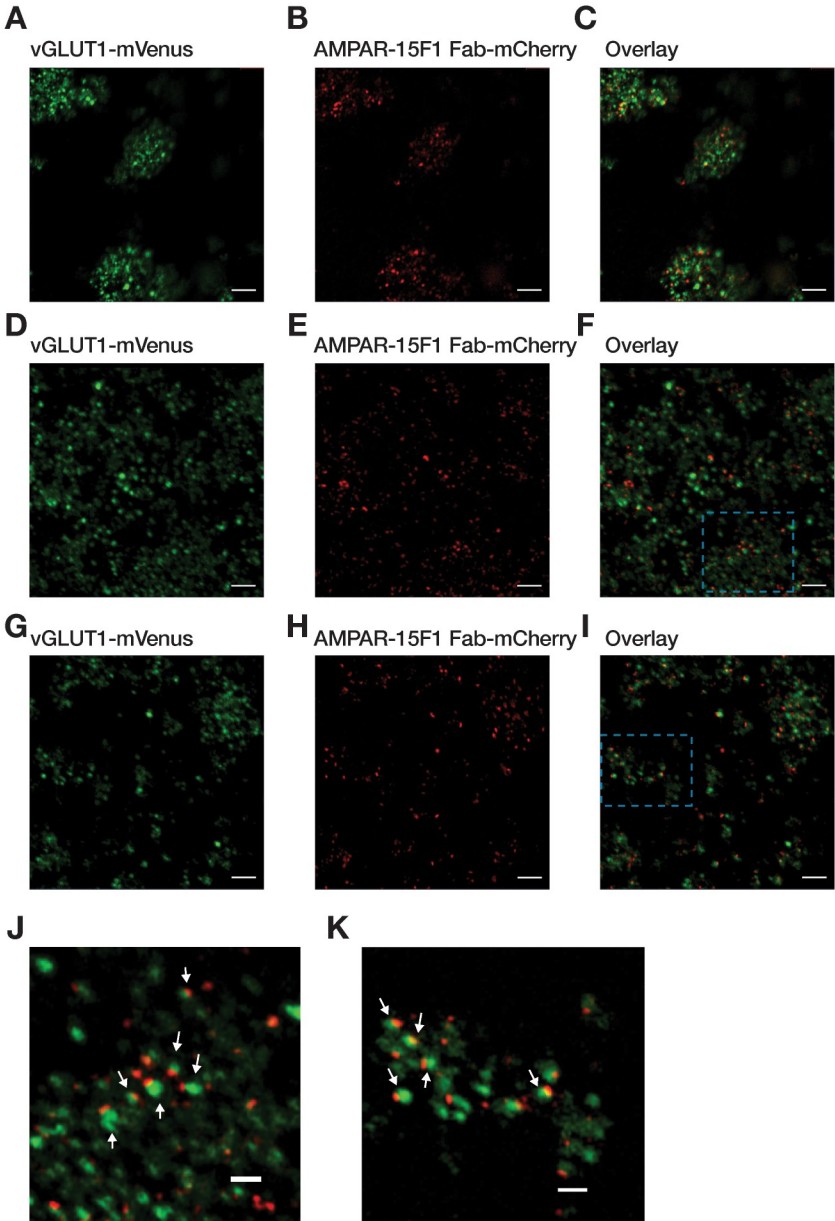

**Fig 3.** Cryo-LSM images of glutamatergic synaptosomes on cryo-EM grids isolated using (A-C) sucrose, (D-F) Ficoll and (G-I) Percoll density gradient centrifugation. Fluorescence signals of vGLUT1-mVenus and AMPAR-15F1 Fab-mCherry at the pre and postsynaptic compartments of synaptosomes are in the green and red channel, respectively. Zoomed-in images of glutamatergic synaptosomes prepared using (J) Ficoll and (K) Percoll density gradient centrifugation corresponding to areas enclosed in cyan dashed box in (F) and (I). Glutamatergic synaptosomes are highlighted with white arrows. Scale bar in (A-I): 5 μm; scale bar in (J,K): 2 μm.

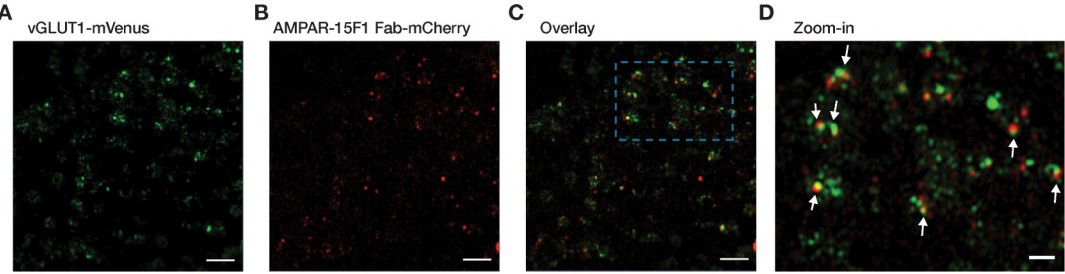

**Fig 4. Cryo-LSM images of glutamatergic synaptosomes on cryo-EM grids passed through 1 μm filter using a thermobarrel extruder.** Fluorescence signals from green, red and both green & red channels corresponding to (A) vGLUT1-mVenus, (B) AMPAR-15F1 Fab-mCherry and (C) both vGLUT1-mVenus & AMPAR-15F1 Fab-mCherry. (D) Zoomed-in area enclosed in cyan dashed box in (C) with glutamatergic synaptosomes highlighted with white arrows. Scale bar in (A-C): 5 μm and scale bar in (D): 2 μm.

density gradient centrifugation. In all instances, the presence of the vGLUT1-mVenus and 15F1 Fab-mCherry bound AMPAR in the synaptosome preparation were indicated by fluorescence signals in the mVenus ($\lambda_{ex}$: 510 nm, $\lambda_{em}$: 535 nm) and mCherry ($\lambda_{ex}$: 580 nm, $\lambda_{em}$: 610 nm) channels, respectively. The elution times of the vGLUT1 (~1850 sec) and AMPA (~1450 sec) correspond to their expected molecular weights, ~175 and ~600 kDa, respectively (Fig 2).

Cryo-EM grids of glutamatergic synaptosomes prepared using a sucrose density gradient had a distinct drawback as compared to synaptosomes prepared using either Ficoll or Percoll density gradient. Synaptosomes prepared using sucrose density gradient tend to form aggregates after application on cryo-EM grids (Fig 3A–3C). A similar event could be observed for undiluted synaptosome samples prepared using Ficoll or Percoll density gradients. However, a 50-fold dilution of the synaptosomes prepared by Ficoll or Percoll density gradient results in a homogeneous distribution on cryo-EM grids (Fig 3D–3I). The presence of glutamatergic synaptosomes is marked by the presence of overlapping green and red fluorescence signals on the EM grids (Fig 3J and 3K).

Further, we filtered the synaptosomes after density gradient centrifugation by passing through a 1 μm filter using a thermobarrel extruder. The filtered synaptosomes were applied on cryo-EM grids and subsequently imaged under cryo-LSM. Glutamatergic synaptosomes subjected to filtration appeared to be more monodisperse, with a uniform distribution (Fig 4A–4D).

Cryo-FIB milling of synaptosomes resulted in lamellae with a thickness range of ~100–200 nm with sample area of ~3–6 μm (Fig 5A–5C). To examine the presence of fluorescent signals associated with glutamatergic synaptosome on the milled lamella, the grids were imaged using cryo-LSM. Interestingly, the fluorescence signal from glutamatergic synaptosomes corresponding to vGLUT1-mVenus and 15F1 Fab-mCherry bound to AMPAR could be detected, indicating the successful preparation of cryo-FIB milled lamellae of glutamatergic synaptosomes (Fig 6A–6F). To further confirm the presence of synaptosome on lamella, cryo-ET imaging was performed on a FIB-milled lamella and the tomogram was reconstructed. The reconstructed tomogram revealed a typical synaptosome (diameter: < 1 μm) with a presynaptic terminal associated to a much smaller postsynaptic compartment separated by a synaptic cleft of ~ 20 nm. The presynaptic and postsynaptic membranes had a smooth and continuous appearance without any visible signs of aggregation (Fig 6G).

The workflow presented here describes the conditions to prepare glutamatergic synaptosomes using density gradient centrifugation followed by preparation of lamellae using cryo-

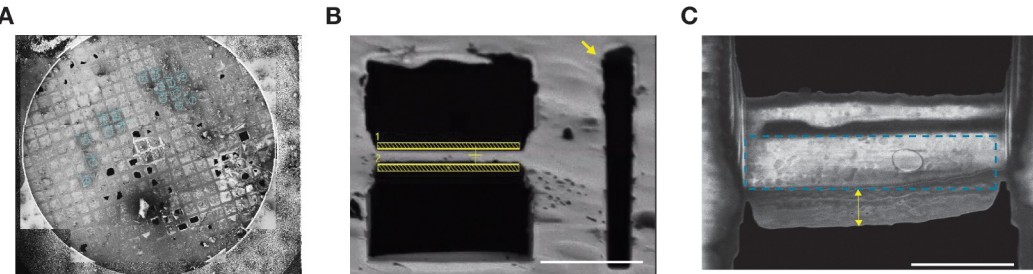

**Fig 5. Cryo-SEM images of synaptosomes at different stages of FIB-milling.** (A) Cryo-SEM images of the whole cryo-EM grid with the selected squares for FIB-milling highlighted with cyan sphere. (B) Representative cryo-SEM image of a lamella preparation during rough milling. The lamella lies between the two milling patterns shown as yellow bars. Lateral micro-expansion joints [58], marked with yellow arrow, are created on both side of the lamella (only shown for the right-hand side) (C) Representative image of a polished synaptosome lamella with the sample area that can be imaged using cryo-ET enclosed in cyan dashed box. The platinum (Pt) gas injection system (GIS) layer is marked with yellow double arrow. Scale bar: 5 μm.

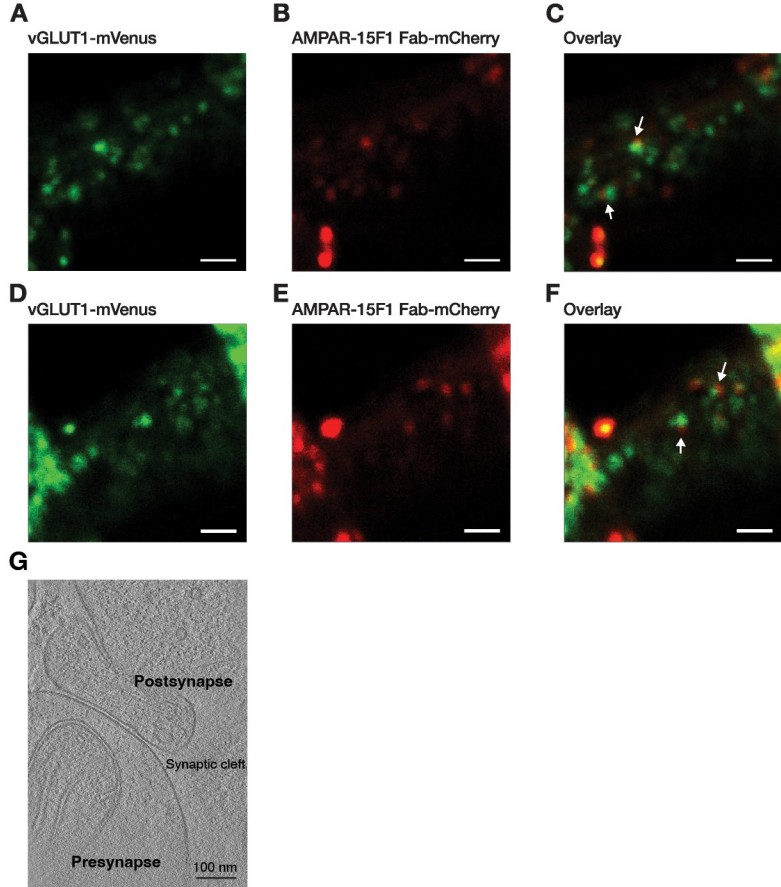

**Fig 6.** Cryo-LSM images of glutamatergic synaptosomes in two cryo-FIB-milled lamellae, (A-C) lamella 1 and (D-F) lamella 2. Fluorescence signal from green, red and both green & red channels corresponding to (A) vGLUT1-mVenus, (B) AMPAR-15F1 Fab-mCherry and (C) overlay of vGLUT1-mVenus and AMPAR-15F1 Fab-mCherry, respectively. Glutamatergic synaptosomes are highlighted with white arrows. Scale bar: 5 μm.

FIB milling suitable for cryo-ET imaging. A comparison was drawn among synaptosomes prepared by sucrose, Ficoll and Percoll to identify the conditions most suitable for cryo-EM grid set up and subsequent lamella preparation. Even though a similar amount of time is required, synaptosomes prepared using sucrose density gradient centrifugation tend to form aggregates on the cryo-EM grids. On the other hand, synaptosomes prepared using Ficoll or Percoll appear to be monodispersed upon dilution. Moreover, synaptosomes isolated by Percoll have an added advantage of involving less preparation time. Using fluorescently labelled vGLUT1 and AMPAR for the pre and postsynaptic compartments aided in distinguishing glutamatergic synaptosomes as well as acted as a guide to identify target areas or glutamatergic synaptosomes during cryo-FIB milling. The ability to obtain the fluorescence signal during cryo-LSM imaging can further be exploited for screening lamellae with glutamatergic synaptosomes and for picking targets after alignment during cryo-ET imaging.

## Supporting information

**S1 Fig. Typical appearance of centrifuge tubes after density gradient centrifugation.** (A) Typical appearance of centrifuge tubes after sucrose density gradient centrifugation. The fraction F2 at the interface between 0.8 and 1.2 M sucrose corresponds to synaptosomes. A similar result is observed for Ficoll density gradient centrifugation with the synaptosome fraction (F2) lying between 8 and 14% Ficoll. In case of sucrose and Ficoll density gradient centrifugation, fractions F1 and F3 represent myelin & membranes and extrasynaptosomal mitochondria, respectively [33,36]. (B) Typical appearance after Percoll density gradient centrifugation. Fraction F4 at the interface between 15 and 23% Percoll corresponds to synaptosomes. Fractions F1, F2, F3 and F5 contain membranes, myelin & membranes, synaptosomes along with membrane vesicles and extrasynaptosomal mitochondria, respectively [38].
(TIF)

**S1 File.**
(PDF)

## Acknowledgments

We thank the Janelia Research Campus for Aquilos cryo-FIB-SEM use, A. Goehring and A. Matsui for maintaining vGLUT1-mVenus mouse colony, A. Matsui for guidance and advice on cryo-LSM, Y. Zhao and J. Yu for providing 15F1 Fab-mCherry plasmid, Y. Zhao for initiating the project, J. Elferich for advice on cryo-EM grid preparation for cryo-FIB milling, R. Hallford for help with manuscript preparation and Gouaux laboratory members for helpful discussions.

## Author Contributions

**Conceptualization:** Eric Gouaux.

**Data curation:** Prerana Gogoi, Momoko Shiozaki.

**Formal analysis:** Prerana Gogoi, Eric Gouaux.

**Funding acquisition:** Eric Gouaux.

**Supervision:** Eric Gouaux.

**Validation:** Prerana Gogoi, Eric Gouaux.

**Visualization:** Prerana Gogoi.

**Writing – original draft:** Prerana Gogoi.

**Writing – review & editing:** Momoko Shiozaki, Eric Gouaux.

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
