## [Decision Letter · Decision Letter 0]

28 Mar 2022

PONE-D-22-00522Isolation, cryo-laser scanning confocal microscope imaging and cryo-FIB milling of mouse glutamatergic synaptosomesPLOS ONE

Dear Dr. Gouaux,

Thank you for submitting your manuscript to PLOS ONE. After careful consideration, we feel that it has merit but does not fully meet PLOS ONE’s publication criteria as it currently stands. Therefore, we invite you to submit a revised version of the manuscript that addresses the points raised during the review process.Please provide a Link, in the Materials and Methods section, to the protocol.io component, using the digital object identifier (DOI) and format provided by protocols.io, as specified in the submissions requirements for Lab Protocols at https://journals.plos.org/plosone/s/submission-guidelines#loc-lab-protocols.

We look forward to receiving your revised manuscript.

Kind regards,

Sang H Lee, Ph.D.

Academic Editor

PLOS ONE

Journal Requirements:

“This work was supported by the NIH (NINDS) grant 2R01NS038631 to E.G. and E.G. is an investigator with the Howard Hughes Medical Institute.”

“EG NIH (NINDS) 2R01NS038631

EG Howard Hughes Medical Institute

The funders had and will not have a role in study design, data collection and analysis, decision to publish, or preparation of the manuscript.”

6. Please include captions for your Supporting Information files at the end of your manuscript, and update any in-text citations to match accordingly. Please see our Supporting Information guidelines for more information: http://journals.plos.org/plosone/s/supporting-information

7. To comply with PLOS ONE submissions requirements, please provide the Protocols.io DOI in the Methods section of the manuscript using this format: “The protocol described in this peer-reviewed article is published on protocols.io, https://dx.doi.org/10.17504/protocols.io[........] and is included for printing as supporting information file 1 with this article.” Please also provide the Protocols.io DOI in the “Protocol DOI” field of the submission form (via “Edit Submission”). For more information, please see our submission guidelines:  https://journals.plos.org/plosone/s/submission-guidelines#loc-guidelines-for-specific-study-types

Reviewers' comments:

Reviewer's Responses to Questions

**Comments to the Author**

1. Does the manuscript report a protocol which is of utility to the research community and adds value to the published literature?

Reviewer #1: Yes

Reviewer #2: No

2. Has the protocol been described in sufficient detail?

Descriptions of methods and reagents contained in the step-by-step protocol should be reported in sufficient detail for another researcher to reproduce all experiments and analyses. The protocol should describe the appropriate controls, sample sizes and replication needed to ensure that the data are robust and reproducible.

Reviewer #1: Yes

Reviewer #2: Partly

3. Does the protocol describe a validated method?

Reviewer #1: No

Reviewer #2: No

4. If the manuscript contains new data, have the authors made this data fully available?

Reviewer #1: Yes

Reviewer #2: N/A

**5. Is the article presented in an intelligible fashion and written in standard English?**

Reviewer #1: Yes

Reviewer #2: Yes

6. Review Comments to the Author

Reviewer #1: The manuscript by Gogoi et al. described the workflow for the preparation of glutamatergic synaptosomes for cryo-electron tomography (cryo-ET) analysis. To identify the glutamatergic synaptosomes, the authors took advantage of the knock-in mouse brain in which presynaptic vGLUT1-mVenus is expressed and the GluA2 specific antibody fragment tagged with mCherry (15F1 Fab-mCherry). After density gradient centrifugation, the authors identified glutamatergic synaptosomes by cryo-confocal fluorescence microscopy, which visualizes the presynaptic vGLUT1-mVenus and postsynaptic AMPAR-15F1 Fab-mCherry-positive synaptosomes. Subsequently, cryo-FIB milling was used for rendering lamellae for cryo-ET analysis. Although cryo-ET studies are now topical and timely issues and the manuscript is clearly written, the actual cryo-ET data (i.e., cryo-ET imaging of this synaptosomes) is missing. So, the readers cannot evaluate this protocol.

Major comments

1. The authors should add the representative cryo-ET images of glutamatergic synaptosomes acquired by this method.

2. The authors should cite the two key references in the cryo-ET studies of synapses and discuss them.

(1) Rubén Fernández-Busnadiego. Cryo-Electron Tomography of the Mammalian Synapse.

Methods Mol Biol. 2018;1847:217-224. doi: 10.1007/978-1-4939-8719-1_16.

(2) Chang-Lu Tao et al. Differentiation and Characterization of Excitatory and Inhibitory Synapses by Cryo-electron Tomography and Correlative Microscopy.

J Neurosci 2018 Feb 7;38(6):1493-1510.doi: 10.1523/JNEUROSCI.1548-17.2017.

Minor comment

The authors should spell out “FSEC” in the line 111 and “Pt GIS” in the lien 161.

Reviewer #2: This brief protocol describes a synaptoneurosome preparation for cryoET studies. The data would be better suited as part of a research article, at present it is unclear whether this workflow will permit structural characterisation of synaptic receptor complexes.

Other comments

1. It would be helpful to present a workflow schematic in the main text figure, as it is currently not described with sufficient clarity.

2. In the “Cryo-FIB milling of glutamatergic synaptosomes” section, fluorescent light microscope images were used for the guidance of FIB milling. An overlay image of FLM and SEM would be needed to demonstrate the alignment.

3. In main text line 138, the authors mentioned filtering of synaptosomes by using a thermobarrel extruder, was this sample used for the following EM grids preparation and FIB milling? Please clarify

4. a TEM image of a representative lamella and an its overlay with the corresponding light microscope image should be added to figure 5.

5. The authors state that by following their procedure the final lamella can reach 100-200 nm thickness, a section of a reconstructed tomogram or other evidence should be added to support this statement.

6. In S1 Fig, the authors mention the existence of other components in density gradient fractions ( “myelin & membranes and extrasynaptosomal mitochondria”). Western blot or other experimental evidence is needed to support the identification of those components.

7. PLOS authors have the option to publish the peer review history of their article (what does this mean?). If published, this will include your full peer review and any attached files.

Reviewer #1: No

Reviewer #2: No

---

## [Author Response · Author response to Decision Letter 0]

12 May 2022

Review Comments to the Author

Reviewer #1: The manuscript by Gogoi et al. described the workflow for the preparation of glutamatergic synaptosomes for cryo-electron tomography (cryo-ET) analysis. To identify the glutamatergic synaptosomes, the authors took advantage of the knock-in mouse brain in which presynaptic vGLUT1-mVenus is expressed and the GluA2 specific antibody fragment tagged with mCherry (15F1 Fab-mCherry). After density gradient centrifugation, the authors identified glutamatergic synaptosomes by cryo-confocal fluorescence microscopy, which visualizes the presynaptic vGLUT1-mVenus and postsynaptic AMPAR-15F1 Fab-mCherry-positive synaptosomes. Subsequently, cryo-FIB milling was used for rendering lamellae for cryo-ET analysis. Although cryo-ET studies are now topical and timely issues and the manuscript is clearly written, the actual cryo-ET data (i.e., cryo-ET imaging of this synaptosomes) is missing. So, the readers cannot evaluate this protocol.

Major comments

1. The authors should add the representative cryo-ET images of glutamatergic synaptosomes acquired by this method.

We thank the reviewer for the suggestion and we have added a reconstructed tomogram of synaptosomes acquired by this method to the revised version of the manuscript.

2. The authors should cite the two key references in the cryo-ET studies of synapses and discuss them.

(1) Rubén Fernández-Busnadiego. Cryo-Electron Tomography of the Mammalian Synapse.

Methods Mol Biol. 2018;1847:217-224. doi: 10.1007/978-1-4939-8719-1_16.

(2) Chang-Lu Tao et al. Differentiation and Characterization of Excitatory and Inhibitory Synapses by Cryo-electron Tomography and Correlative Microscopy.

J Neurosci 2018 Feb 7;38(6):1493-1510.doi: 10.1523/JNEUROSCI.1548-17.2017.

As suggested by the reviewer, both the references are discussed in the revised version of the manuscript as suggested by the reviewer.

Minor comment

The authors should spell out “FSEC” in the line 111 and “Pt GIS” in the lien 161.

“FSEC” and “Pt GIS” have been spelled out in the revised version of the manuscript. 

Reviewer #2: This brief protocol describes a synaptoneurosome preparation for cryoET studies. The data would be better suited as part of a research article, at present it is unclear whether this workflow will permit structural characterisation of synaptic receptor complexes.

Other comments

1. It would be helpful to present a workflow schematic in the main text figure, as it is currently not described with sufficient clarity.

We thank the reviewer for the suggestion and we have incorporated a workflow schematic in the main text (Fig 1).

2. In the “Cryo-FIB milling of glutamatergic synaptosomes” section, fluorescent light microscope images were used for the guidance of FIB milling. An overlay image of FLM and SEM would be needed to demonstrate the alignment.

Although fluorescence light images were used to guide the FIB-milling, we were unable to discern synaptosome-like features in the SEM images, simply due to the constraints of the imaging mode. Thus, we do not believe that an overlay of the FLM and SEM images would constructively augment the manuscript. What we did do, however, was to perform fluorescence imaging on the FIB-milled lamella, to confirm that our FLM guided milling was successful, as evidenced by the presence of adjacent or overlapping red and green fluorescence (Fig 6).

3. In main text line 138, the authors mentioned filtering of synaptosomes by using a thermobarrel extruder, was this sample used for the following EM grids preparation and FIB milling? Please clarify

In the revised version of the manuscript, we have mentioned that the filtered synaptosomes were used for the preparation of EM grids.

4. a TEM image of a representative lamella and an its overlay with the corresponding light microscope image should be added to figure 5.

TEM images were acquired on the lamella for which overlapping green and red fluorescence signal were detected in the cryo-LSM. However, we have not employed any tools or other ways to overlay the images.

5. The authors state that by following their procedure the final lamella can reach 100-200 nm thickness, a section of a reconstructed tomogram or other evidence should be added to support this statement.

The thickness of the lamella was measured during cryo FIB milling using the AutoTEM 2.0 software and monitored using SEM. To further support the preparation of successful synaptosome lamellae, we have added a reconstructed tomogram to the revised version of the manuscript. 

6. In S1 Fig, the authors mention the existence of other components in density gradient fractions ( “myelin & membranes and extrasynaptosomal mitochondria”). Western blot or other experimental evidence is needed to support the identification of those components.

In this study, the steps for density gradient separation of synaptosomes from other components has been performed and similar results were reproduced as described in previous studies (Gray and Whittaker, 1962; Cotman and Matthews, 1971; Dunkley et al., 2008). All these studies have been cited appropriately (S1 Fig legend). Given the extensive prior documentation, we respectively assert that further analysis is not a productive use of time or resources.

---

## [Decision Letter · Decision Letter 1]

8 Jul 2022

Isolation, cryo-laser scanning confocal microscope imaging and cryo-FIB milling of mouse glutamatergic synaptosomes

PONE-D-22-00522R1

Dear Dr. Gouaux,

We’re pleased to inform you that your manuscript has been judged scientifically suitable for publication and will be formally accepted for publication once it meets all outstanding technical requirements.

Kind regards,

Sang H Lee, Ph.D.

Academic Editor

PLOS ONE

Additional Editor Comments (optional):

Reviewers' comments:

Reviewer's Responses to Questions

**Comments to the Author**

1. Does the manuscript report a protocol which is of utility to the research community and adds value to the published literature?

Reviewer #1: Yes

2. Has the protocol been described in sufficient detail?

Descriptions of methods and reagents contained in the step-by-step protocol should be reported in sufficient detail for another researcher to reproduce all experiments and analyses. The protocol should describe the appropriate controls, sample sizes and replication needed to ensure that the data are robust and reproducible.

Reviewer #1: Yes

3. Does the protocol describe a validated method?

Reviewer #1: Yes

4. If the manuscript contains new data, have the authors made this data fully available?

Reviewer #1: Yes

**5. Is the article presented in an intelligible fashion and written in standard English?**

Reviewer #1: Yes

6. Review Comments to the Author

Reviewer #1: The authors suitably responded to all my comments. Now, the paper will provide readers with a useful workflow for the preparation glutamatergic synaptosomes for Cryo-ET analysis.

7. PLOS authors have the option to publish the peer review history of their article (what does this mean?). If published, this will include your full peer review and any attached files.

Reviewer #1: No

---

## [Editor Report · Acceptance letter]

4 Aug 2022

PONE-D-22-00522R1 

Isolation, cryo-laser scanning confocal microscope imaging and cryo-FIB milling of mouse glutamatergic synaptosomes 

Dear Dr. Gouaux:

I'm pleased to inform you that your manuscript has been deemed suitable for publication in PLOS ONE. Congratulations! Your manuscript is now with our production department. 

Kind regards, 

on behalf of

Dr. Sang H Lee 

Academic Editor

PLOS ONE